# INCREMENTAL TRAINING OF MULTI-GENERATIVE ADVERSARIAL NETWORKS

## ABSTRACT

Generative neural networks map a standard, probability distribution to a complex high-dimensional distribution, which represents the real world data set. However, a determinate input distribution as well as a specific architecture of neural networks may impose limitations on capturing the diversity in the high dimensional target space. To resolve this difficulty, we propose a training framework that greedily produce a series of generative adversarial networks that incrementally capture the diversity of the target space. We show theoretically and empirically that our training algorithm converges to the theoretically optimal distribution, the projection of the real distribution onto the convex hull of the network's distribution space.

## 1 INTRODUCTION

Generative Adversarial Nets (GAN) Goodfellow et al. (2014) is a framework of estimating generative models. The main idea Goodfellow (2017) is to train two target network models simultaneously, in which one, called the generator, aims to generate samples that resemble those from the data distribution, while the other, called the discriminator, aims to distinguish the samples by the generator from the real data. Naturally, this type of training framework admits a nice interpretation as a two-person zero-sum game and interesting game theoretical properties, such as uniqueness of the optimal solution, have been derived Goodfellow et al. (2014). It is further proved that such adversarial process minimizes certain divergences, such as Shannon divergence, between the generated distribution and the data distribution.

Simply put, the goal of training a GAN is to search for a distribution in the range of the generator that best approximates the data distribution. The range is often defined by the input latent variable $\mathbf{z}$ and its specific architecture, i.e., $\Pi = \{G(\mathbf{z}, \boldsymbol{\theta}), \boldsymbol{\theta} \in \boldsymbol{\Theta}\}$. When the range is general enough, one could possibly find the real data distribution. However, in practice, the range is usually insufficient to perfectly describe the real data, which is typically of high dimension. As a result, what we search for is in fact the I-projection Csiszár & Shields (2004) of the real data distribution on $\Pi$.

Consider two cases:

1. The range of the generator $\Pi$ is convex (see figure 1(a)), or it is not convex but the projection of the real data distribution on $\Pi$'s convex hull (CONV $\Pi$) is in $\Pi$ (see figure 1(b)).
2. The range of the generator is non-convex and the projection of the real data distribution in CONV $\Pi$ is not in the range $\Pi$ (see figure 1(c)).

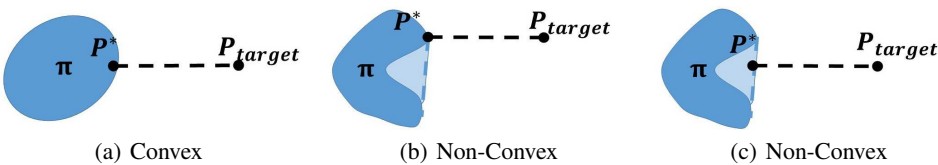

(a) Convex          (b) Non-Convex          (c) Non-Convex

Figure 1: Different cases of the distribution projection on $\Pi$'s convex hull

In case 1, one can find the optimal distribution in $\Pi$ to approximate real data set in CONV $\Pi$. But in case 2, using standard GANs with a single generator, one can only find the distribution in $\Pi$

that is nearest to the projection. It then makes sense to train multiple generators and use a convex combination of them to better approximate the data distribution (than using a single generator in the non-convex case (see figure 1(c))).

The above argument is based on the assumption that one could achieve global optimality by training, while this is not the case in general. When reaching a local optimal distribution, in order to improve performance, do we need to add more generators and restart training? In this paper, we put forward a sequential training procedure that adds generators one by one to improve the performance, without retraining the previously added generators. Our contributions can be summarized as follows.

- We derive an objective function tailored for such a incremental training process. The objective function takes both the real data distribution and the pre-learned distribution into consideration. We show that with this new objective, we actually maximize marginal contribution when adding a new generator. We also put forward an incremental training algorithm based on the new objective function.
- We prove that our algorithm always converges to the projection of real data distribution to the convex hull of the ranges of generators, which is the optimal solution with multiple generators. This property continues to hold in online settings where target distribution changes dynamically.
- Our experiments show that our algorithm can overcome the local optimal issue mentioned above. We perform experiments on a synthetic dataset as well as two real world datasets, e.g., CelebA and MNIST, and conclude that our algorithm could improve the mixture distribution even in the case where the range is not sufficient enough.
- Experiments also show that, compared with previous methods, our algorithm is fast and stable in reducing the divergence between mixture distribution and the real data.

## 1.1 RELATED WORKS

Recently, there have been intensive researches on improving the performance of generative adversarial neural networks. Two lines of works are closely related to our paper. They focus mainly on improving the discriminator and the generator respectively.

### 1.1.1 IMPROVING THE DISCRIMINATOR

The Unrolled GAN introduced by Metz et al. (2016) improves the discriminator by unrolling optimizing the objective during training, which stabilizes training and effectively reduces the mode collapse. D2GAN proposed by Nguyen et al. (2017) utilizes two discriminators to minimize the KL-divergence and the reverse KL-divergence respectively. It treats different modes more fairly, and thus avoids mode collapse. DFM introduced by Warde-Farley & Bengio (2016) brings a Denoising AutoEncoder (DAE) into the generator's objective to minimize the reconstruction error in order to get more information from the target manifold. Mroueh et al. (2017) proposed McGan based on mean and covariance feature matching to stabilize the training of GANs. Finally, WGAN introduced by Arjovsky et al. (2017) employs the Wasserstein distance, which is a more appropriate measure of performance, and achieves more stable performance.

These works are different from ours since they focus on the discriminator by measuring the divergence between the generated data and the real data more precisely. However, our work fixes the discriminator and tries to enrich the expressiveness of the generator by combining multiple generators.

### 1.1.2 IMPROVING THE GENERATOR

Wang et al. (2016) proposes two methods to improve the training process. The first is self-ensembling GANs, which assembles the generators from different epochs to stabilize training. The other is Cascade GAN, where the authors train new generator using the data points with highest values from the discriminator. These two methods are heuristic ways to improve training, but with no theoretical guarantee.

Hoang et al. (2017) and Ghosh et al. (2017) proposed the methods called MGAN and multi-agent GANs respectively. The former introduces a classifier into the discriminator to catch different

modes, while the later employ a new component into the generators' objective to promote diversity. Arora et al. (2017) introduces a new metric on distributions and proposes a MIX+GAN to search for an equilibrium. But all these methods need to train the multiple generators simultaneously, and none of them can deal with the case when the training process reaches a local optima. Also, these models lack flexibility, in the sense that when one tries to change the number of generators, all the generators need to be retrained.

Another closely related work is Tolstikhin et al. (2017), in which the authors propose a method called AdaGAN, which is based on a robust reweighting scheme on the data set inspired from boosting. The idea is that the new generators should focus more on the previous bad training data. But AdaGAN and other boosting-like algorithms are based on the assumption that one generator could catch some modes precisely, which may not be reasonable since the generator always learns to generate the average samples among the real data set in order to obtain low divergence, especially when the generator's range is under condition of Figure 1(c). In Section 5, we compare our algorithm with AdaGAN with different dataset.

## 2 PRELIMINARIES

A GAN Goodfellow et al. (2014) takes samples (a.k.a. latent variables $\mathbf{z}$) from a simple and standard distribution as its input and generates samples in a high dimensional space to approximate the target distribution. This is done by training a generative neural network and an auxiliary discriminative neural network alternatively. An f-GAN Nowozin et al. (2016) generalizes the adversarial training as minimizing the f-divergence between the real data distribution and the generated distribution, defined as $D_f(p\|q) = \int_{\mathbf{x}} q(\mathbf{x}) f\left(\frac{p(\mathbf{x})}{q(\mathbf{x})}\right) \, d\mathbf{x}$. A GAN is a special f-GAN that minimizes the Jensen-Shannon divergence. The general objective function of an f-GAN can be defined as follows:

$$\min_{\boldsymbol{\theta}} \max_{\boldsymbol{\xi}} F(\boldsymbol{\theta}, \boldsymbol{\xi}) = E_{\mathbf{x} \sim p_{data}}[T_{\boldsymbol{\xi}}(\mathbf{x})] + E_{\mathbf{x} \sim p_{\boldsymbol{\theta}}}[-f^*(T_{\boldsymbol{\xi}}(\mathbf{x}))].$$

Here $f^*$ is the conjugate function of $f$ in f-divergence; T represents a neural network regarded as the corresponding discriminator; finally, $\boldsymbol{\theta}$ and $\boldsymbol{\xi}$ denote the parameters of the generator and the discriminator, respectively.

### 2.1 GENERATOR GROUP

The adversarial training method proposed by Goodfellow et al. (2014) is playing a minimax game between the generator and the discriminator. Such a method can be caught in local optima and thus is undesirable (e.g., mode collapse).

In this paper we propose a novel framework to train multiple generators sequentially: We maintain a group of generators (empty at the beginning) as well as their corresponding weights, then add new generators into the group one by one and rebalance the weights. In particular, only the newly added generator at each step is trained. The purpose here is to augment the capacity of the group of generators and mitigate the local optima issue.

Define the distribution range of a generator as $\Pi = \{p \mid p = G(\mathbf{z}, \boldsymbol{\theta}), \boldsymbol{\theta} \in \boldsymbol{\Theta}\}$, i.e., the set of distributions that the generator can produce with different parameter $\theta$. The distribution range is determined by the distribution of input $\mathbf{z}$ and the architecture of the generative network. Define a generator group as $G = \{G_1, G_2, \ldots, G_n\}$, where $G_i$ is the generator added in step $i$. We associate each generator with a weight $\omega_i > 0$. Then the mixed distribution of the group is: $p_{pre}(\mathbf{x}) = \frac{1}{\Phi_n} \sum_{i=1}^{n} \omega_i p_i(\mathbf{x})$, where $p_i(\mathbf{x}) = G_i(\mathbf{z})$ and $\Phi_n = \sum_{i=1}^{n} \omega_i$ is the sum of weights. When a new generator $G_{n+1}$ joins in the group $G$, the group becomes $G' = G \cup \{G_{n+1}\}$ and the mixed distribution becomes

$$p_{now}(\mathbf{x}) = \frac{1}{\Phi_{n+1}} \sum_{i=1}^{n+1} \omega_i p_i(\mathbf{x}) = \frac{1}{\Phi_{n+1}} \left( \Phi_n \cdot p_{pre}(\mathbf{x}) + \omega_{n+1} \cdot p_{n+1}(\mathbf{x}) \right).$$

## 3 AN INCREMENTAL TRAINING FRAMEWORK

In this section, we describe how we use a generator group to improve the performance and tackle the local optima issue mentioned previously. To train such a generator group, we propose an incremental

training algorithm (algorithm 1) adding generators to the group sequentially. In algorithm 1, we use

---

**Algorithm 1** Incremental Training Algorithm

---

**Predetermined:** $p_{real}$, $p_{\mathbf{z}}$, $G_i(\mathbf{z})$, $i \in \{1, 2, \ldots, N\}$ $\omega_i$, $i \in \{1, 2, \ldots, N+1\}$
**Input** $p_{real}$, $p_{\mathbf{z}}$, $\omega_i$, $i > 0$.
**Output** generator group $G$.
Initialize $i \leftarrow 1$.
**repeat**
    Build and initialize generator $G_i$ using the same network structure.
    Set target distribution for $G_i$ to be $p_{target} = \frac{\Phi_i \cdot p_{real} - \sum_{j<i} \omega_j p_j}{\omega_i}$.
    Train $G_i$ to minimize $D(p_{target}, p_i)$.
    $i \leftarrow i + 1$.
**until** Convergence

---

$D(\cdot, \cdot)$ to denote the "distance" between two distributions, which can be any divergence (e.g., f-divergence or Wasserstein distance) or a general norm.

The key step in algorithm 1 is the choice of the target distribution for training $G_i$. Ideally, if $D(p_{target}, p_i) = 0$, then we have $p_i = p_{target}$. In this case, $p_{real} = \frac{1}{\Phi_i} \sum_{j=1}^{i} \omega_j p_j$ and after adding $G_i$, the generator group $G$ can perfectly produce the desired distribution $p_{real}$. However, in general, we have $D(p_{target}, p_i) \neq 0$ and our algorithm proceeds in a greedy fashion, i.e., it always maximizes the marginal contribution of $G_i$ to the generator group $G$. We devote the rest of this section to proving the above statement.

### 3.1 MARGINAL CONTRIBUTION MAXIMIZATION AT EACH ROUND

In algorithm 1, we use different loss functions for each generators. The marginal contribution of the $(N+1)$-th generator is as follows when we adopt f-divergence as the distance measure:

$$V(G_{N+1}) = D_f(p_{pre}||p_{real}) - D_f(p_{now}||p_{real}) = \int_{\mathbf{x}} p_{real} \left[ f\left(\frac{p_{pre}}{p_{real}}\right) - f\left(\frac{p_{now}}{p_{real}}\right) \right] \mathrm{d}\mathbf{x} \quad (1)$$

To get a better approximation to the real distribution, we fix the existing generators in the group and tune the parameters of the new generator to minimize the distance between the new group and the real distribution. In fact, this is equivalent to maximizing the marginal contribution of the new generator $V(G_{N+1})$ by selecting $G_{N+1} = arg\,max_{G'_{N+1}} V(G'_{N+1})$. Formally,

**Proposition 1.** *For any distribution $p_{real}$ and any existing generator group $G = (G_1, G_2, \ldots, G_N)$, the optimal target distribution $p^*_{N+1}$ for a new generator $G_{N+1}$ to join the group $G$ is*

$$p^*_{N+1}(\mathbf{x}) = \left[ \frac{\Phi_{N+1} \cdot p_{real}(\mathbf{x}) - \Phi_N \cdot p_{pre}(\mathbf{x})}{\omega_{N+1}} \right]^+, \; where[\cdot]^+ = \max\{\cdot, 0\}$$

.

To show this, we first introduce the $\chi^2$-divergence.

**Definition 1** ($\chi^2$-divergence). *The $\chi^2$-divergence between distribution $p$ and $q$ is:* $D_{\chi^2}(p||q) = \int_{\mathbf{x}} \frac{(p(\mathbf{x}) - q(\mathbf{x}))^2}{q(\mathbf{x})} d\mathbf{x}$. *Note that $\chi^2$-divergence is a special case of the f-divergence:* $D_{\chi^2}(p||q) = D_f(p||q)$, *when* $f(\mathbf{u}) = (\mathbf{u} - \mathbf{1})^2$

In fact, with some mild assumptions on $f$, the $f$-divergence is well-approximated by $\chi^2$-divergence when $p$ and $q$ are close. The following lemma can be obtained via Taylor expansion Csiszár & Shields (2004).

**Lemma 1.** *For any f-divergence with $f(u)$, if $f(u)$ is twice differentiable at $u = 1$ and $f''(1) > 0$, then for any $q$ and $p$ close to $q$ we have:* $D_f(p||q) \sim \frac{f''(1)}{2} D_{\chi^2}(p||q)$.

*Proof of proposition 1.* We rewrite the objective function equation 1 for $\chi^2$-divergence:

$$V(G_{N+1}) = \int_{\mathbf{x}} p_{real} \left[ f\left(\frac{p_{pre}}{p_{real}}\right) - f\left(\frac{p_{now}}{p_{real}}\right) \right] \mathrm{d}\mathbf{x} = \int_{\mathbf{x}} \frac{(p_{real} - p_{pre})^2 - (p_{real} - p_{now})^2}{p_{real}} \mathrm{d}\mathbf{x}.$$

**Algorithm 2** Training $G_{N+1}$ by Gradient Descent

---

**Input:** $p_{real}, p_z, \eta, \omega_{N+1}$ and $G_i(\cdot, \boldsymbol{\theta}_i), \omega_i$, $i \in \{1, 2, \ldots, N\}$
**repeat**
    Sample $\{\boldsymbol{x}_{real}^j\}$ from $p_{real}$
    Sample $\{\boldsymbol{z}_j\}$ from $p_{\mathbf{z}}$
    **for** $i = 1$ **to** $N$ **do**
        Obtain samples by $G_i(\boldsymbol{z}_j, \boldsymbol{\theta}_i)$ as $\{\boldsymbol{x}_{G_i}^j\}$
        in Group G
    **end for**
    Generate samples $\{\boldsymbol{x}_{gen}^j\}$ by $G_{N+1}$
    Update $\boldsymbol{\xi}^{t+1} = \boldsymbol{\xi}^t + \eta \nabla_{\boldsymbol{\xi}} F(\boldsymbol{\theta}_{N+1}^t, \boldsymbol{\xi}^t)$
    Update $\boldsymbol{\theta}_{N+1}^{t+1} = \boldsymbol{\theta}_{N+1}^t - \eta \nabla_{\boldsymbol{\theta}} F(\boldsymbol{\theta}_{N+1}^t, \boldsymbol{\xi}^t)$
**until** Convergence
**Output:** $G_{N+1}(\cdot, \boldsymbol{\theta}_{N+1})$

---

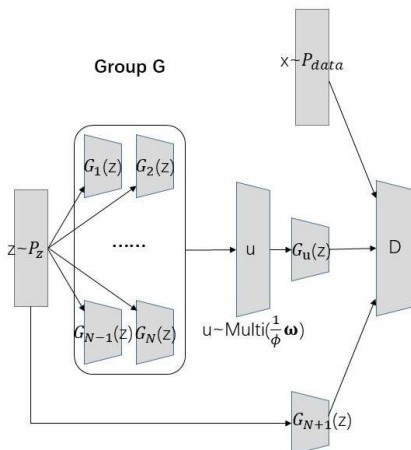

Figure 2: A framework for training $G_{N+1}$

Based on the former definition, we obtain $V(G_{N+1}) = \int_{\mathbf{x}} \frac{\omega_{N+1} \cdot [-\omega_{N+1} \cdot p_{N+1}^2 + A \cdot p_{N+1} + B]}{(\Phi_{N+1})^2 p_{real}} \, d\mathbf{x}$, where $A = 2(\Phi_{N+1} \cdot p_{real} - \Phi_N \cdot p_{pre})$ and $B = (\Phi_{N+1} + \Phi_N) p_{pre}^2 - 2\Phi_{N+1} \cdot p_{pre} \cdot p_{real}$.
To maximize the quadratic function $V(G_{N+1})$, we have $p_{N+1}^* = [A/2\omega_{N+1}]^+$, which concludes the proof. $\qquad\square$

### 3.2 Algorithms to Train $G_i$

According to algorithm 1, in each round, a new generator $G_{N+1}$ is added and the loss function is set to be $D(p_{target}, p_{N+1})$. Therefore, when training each generator $G_i$, the target distribution only depends on the real distribution and the previous generators in $G$. In particular, both of them are already known (figure 2).

To minimize $D(p_{target}, p_{G_{N+1}})$, we conduct adversarial training by using an auxiliary discriminator T:

$$F(\boldsymbol{\theta}, \boldsymbol{\xi}) = E_{\mathbf{x} \sim p_{target}}[T_{\boldsymbol{\xi}}(\mathbf{x})] + E_{\mathbf{x} \sim p_{\boldsymbol{\theta}}}[-f^*(T_{\boldsymbol{\xi}}(\mathbf{x}))],$$

where by the linearity of expectation:

$$E_{\mathbf{x} \sim p_{target}}[T_{\boldsymbol{\xi}}(\mathbf{x})] = \frac{\{\Phi_{N+1} E_{\mathbf{x} \sim p_{real}}[T_{\boldsymbol{\xi}}(\mathbf{x})] - \Phi_N E_{\mathbf{x} \sim p_{pre}}[T_{\boldsymbol{\xi}}(\mathbf{x})]\}}{\omega_{N+1}}$$

$$= \frac{\{\Phi_{N+1} E_{\mathbf{x} \sim p_{real}}[T_{\boldsymbol{\xi}}(\mathbf{x})] - \sum_{i=1}^N \omega_i \cdot E_{\mathbf{x} \sim p_i}[T_{\boldsymbol{\xi}}(\mathbf{x})]\}}{\omega_{N+1}}.$$

Based on these, we propose an incremental training algorithm for $G_{N+1}$ as algorithm 2.

## 4 Theoretical Analyses

In this section, we show that although our framework which trains each generator in a greedy way, the output distribution of the generator group will always converge. Furthermore, the converged distribution is the closest one to the target distribution among the set of all possible distributions that a group of generators can produce (i.e., the optimal one within the distribution range of the group of generators).

Recall our notation that the distribution range of a generator is $\Pi$. By taking a convex combination of multiple generators (with the same architecture), the set of all possible output distributions becomes the convex hull of $\Pi$: CONV $\Pi = \{p(\mathbf{x}) \mid p(\mathbf{x}) = \frac{1}{\Phi_N} \sum_{i=1}^N \omega_i p_i(\mathbf{x})\}$, where $\omega_i > 0$ is the weight of the $i$-th generator, $p_i(\mathbf{x}) = G_i(\mathbf{z}), 1 \le i \le N$, and $G_i$ represent the generative neural networks with the same architecture but different parameters. One can consider the parameters $\frac{\omega_i}{\Phi_N}$ as the probability of choosing the $i$-th generator as the final output.

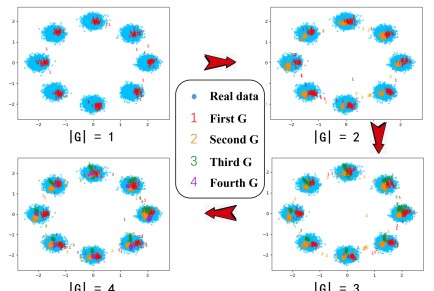

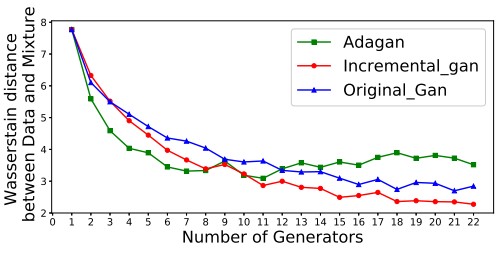

Figure 3: We train 4 generators to catch the modes (i.e., 8 Gaussian distributions).

Figure 4: Wasserstein distance between $p_{real}$ and $p_{now}$ as the number of generators increases.

### 4.1 CONVERGENCE ANALYSIS

Our algorithm greedily optimizes each $G_{N+1}$ to minimize $D(p_{target}, p_{G_{N+1}})$. By the Pinsker's inequality, the total variation distance between $p_{target}$ and $p_{G_{N+1}}$ is upper bounded by $\sqrt{D(p_{target}, p_{G_{N+1}})/2}$ and we can easily extend it to $\chi^2$-divergence by $D_{KL}(p||q) \leq D_{\chi^2}(p||q) + 0.42$. In other words, while greedily optimizing each $G_{N+1}$, the distance between $p_{target}$ and $p_{G_{N+1}}$ is also *approximately* minimized. Hence it is reasonable to assume that for each $G_{N+1}$, its distance to $p_{target}$ is approximately minimized with some tolerance $\epsilon \geq 0$, i.e., $\|p_{G_{N+1}} - p_{target}\| \leq \inf \|p - p_{target}\| + \epsilon$. Under such an assumption, our algorithm approximately converges to the the optimal distribution in CONV $\Pi$:

**Proposition 2.** *For any $\Pi$ that is connected and bounded, algorithm 2 approximately converges to the optimal distribution within the closure of the convex hull $\overline{\text{CONV } \Pi}$ of $\Pi$.*

To simplify the argument, we fix each $\omega_i$ to be 1 and embed the discrete probability distributions into a Hilbert space. In this case, each $G_{N+1}$ approximately minimizes the distance to $p_{target} = (N+1)p_{real} - \sum_{i=1}^{N} p_{G_i}$ can be formalized as:

$$\|p_{G_{N+1}} - p_{target}\| \leq \inf_{p \in \Pi} \|p - p_{target}\| + \epsilon,$$

and our algorithm approximately converges to the optimal distribution in $\overline{\text{CONV } \Pi}$ if as $N \to \infty$,

$$\|\frac{1}{N}\sum_{i=1}^{N} p_{G_N} - p_{real}\| \leq \inf_{p \in \overline{\text{CONV } \Pi}} \|p - p_{real}\| + \epsilon.$$

Then proposition 2 is implied by the following lemma.

**Lemma 2.** *Consider a connected and bounded subset $\Pi$ of a Hilbert space $\mathcal{H}$ and any target $\rho \in \mathcal{H}$. Let $\{p_n^*\}_{n=1}^{\infty}$ be a sequence of points in $\Pi$ such that for $\rho_{target} = (n+1)\rho - nT_n$,*

$$\|p_{n+1}^* - \rho_{target}\| \leq \inf_{p \in \Pi} \|p - \rho_{target}\| + \epsilon, \qquad (\star)$$

*where $T_n = \frac{1}{n}\sum_{i=1}^{n} p_n^*$ and $\epsilon \geq 0$ is a constant. Then for any $\delta > 0$, there exists $N > 0$ such that*

$$\forall n > N, \|T_n - \rho\| \leq \inf_{p \in \overline{\text{CONV } \Pi}} \|p - \rho\| + (1 + \delta)\epsilon.$$

**Corollary 1.** *With the finite change of target distribution, algorithm 2 can converge to the new optimal distribution within $\overline{\text{CONV } \Pi}$.*

Due to the space limit, we send the proof to the appendix.

Based on corollary 1, regardless the change of target distribution, as long as it is an finite variation, algorithm 2 can converge to the projection of new target distribution. Due to the sequential nature and the above theoretical guarantee, our algorithm naturally generalizes the dynamic online settings.

## 5 EXPERIMENTS

We test our algorithm on a synthesized Gaussian distribution dataset and two well-known real world datasets: CelebA and MNIST, which are the complex high dimensional real world distributions.

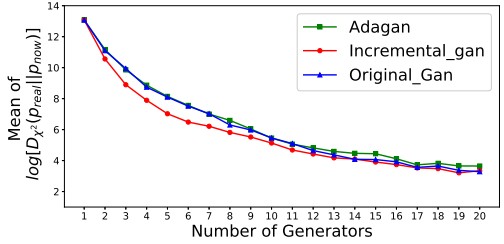 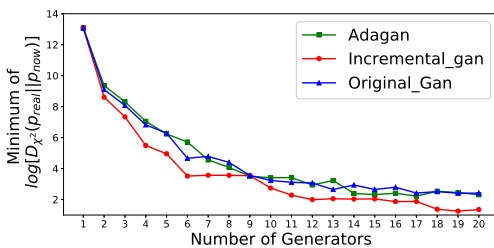

Figure 5: The mean $log(D_{\chi^2}(p_{real}||p_{now}))$ of 30 repetition experiments .

Figure 6: The minimum $log(D_{\chi^2}(p_{real}||p_{now}))$ of 30 repetition experiments .

We design the experiment to test our sequential training algorithm. The main purpose is not to demonstrate high quality results, e.g., high definition pictures, but to show that our algorithm can search for an appropriate distribution that significantly improved the performance of mixture distributions as the number of generators increase, especially when the generator's range is rather limited. In all experiments, we use the Adam optimizer Kingma & Ba (2014) with learning rate of $5 \times 10^{-5}$, and $\beta_1 = 0.5$, $\beta_2 = 0.9$. Finally, we set weights $\omega_i = 1$ for convenience.

**Metric.** As the method mentioned in Nowozin et al. (2016), when we fix the generator, we can train an auxiliary neural network to maximize the derived lower bound to measure the divergence between the generated data and the real data, i.e., $D_f(P||Q)$. Based on these theories, we import an auxiliary neural network to measure the performance of different methods. The architecture of the auxiliary neural network is the same as the discriminator used in each experiment. We train it for 50 epochs, which is enough to measure the differences. Then we take the mean value of the last 100 iterations as the final output.

**Synthesized data.** In this part, we design some experiments in $R^2$ space. The dataset is sampled from 8 independent two-dimensional Gaussian distributions (i.e., the blue points in figure 3). The model is previously proposed by Metz et al. (2016).

Firstly, following the experiment designed in Metz et al. (2016) and Hoang et al. (2017), we choose the latent variable $\mathbf{z}$ in a high dimensional space as $\mathbf{z} \sim N(0, I_{256})$, i.e., the distribution projection is likely to be in the generator's range, which meets the condition of figure 1(a) or figure 1(b). In figure 3, the blue points are the real data while the corresponding colored number represents the data points generated by each generator respectively. As figure 3 shows, we train up to 4 generators to approximate the data distribution and the first generator tends to catch the data with high probability around the centre of each Gaussian. As the number of generators increasing, generated data tends to cover the data away from the centre in order to be complementary to previous mixture distributions and thus gains a considerable marginal profit. These results demonstrate our marginal maximization algorithm can promote the mixture distributions to cover the data with low probabilities.

Secondly, we reduce the dimension of $z$ to 1, i.e., $z \sim N(0, 1)$ and simplify the corresponding network architecture, so that the condition of figure 1(c) is likely met. In this part, we compare our algorithm with the state of the art incremental training method AdaGAN Tolstikhin et al. (2017) and the baseline method Orignal GAN.[1] We train up to 20 generators in each experiment with the same starting generator (i.e., identical first generator for each method), then measure the $D_{\chi^2}(p||q)$ between real distribution and the generated mixed distribution. We repeat the experiment for 30 times to reduce the effect of random noises. Figure 5 and figure 6 illustrate the average and the best performance with different numbers of generators, respectively. According to the result, our algorithm approaches to $p_{real}$ faster than the other two methods and achieves the best performance among all three methods.

In summary, our algorithm outperforms the other two both in terms of the speed of converging to the real distribution and the quality of the final result under the case of figure 1(c).

**MNIST.** In this experiment, we run our algorithm on the MNIST dataset LeCun et al. (1998). We design this experiment to measure the performance of our algorithm for a more complex data distribution. We choose the latent variable as $z \sim N(0, 1)$ to limit the corresponding generator range and

---

[1]The only difference is the target distribution, for Original GAN, we train each generator with the real data distribution, i.e., $p_{real}$, then make a convex combination of them.

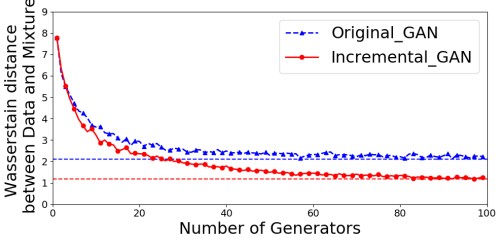 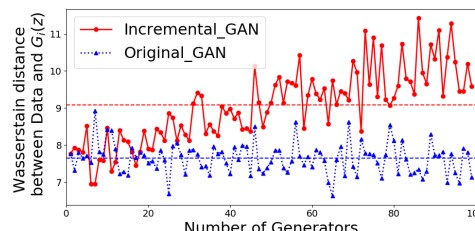

Figure 7: W-distance between mixture distribution and the real data.

Figure 8: W-distance between distribution $G_i(\mathbf{z}, \boldsymbol{\theta})$ and the real data .

use the Wasserstein distance to measure the gap between mixed distribution and the real dataset.[2] Then we train up to 22 generators to approximate the real distribution and the result is showed in figure 4. Our algorithm outperforms the Original GAN but is inferior to the AdaGAN with the first 8 generators. As the number of generators increases, AdaGAN seems to run into a bottleneck while both our algorithm and the Original GAN gradually approximate to the real data distribution.

In order to analysis the convergence, we further train up to 100 generators with both our algorithm and the original GAN. In figure 7, the horizontal dash lines represent the minimum value of the Wasserstein distance for the two method respectively. As showed in figure, the distance gradually decrease with the number of generators increasing, and our algorithm is much faster to reduce the distance and can even obtain a better performance. More over, as we tends to investigate the property of each generator in the generators' group, we measure the Wasserstein distance between distribution $G(\mathbf{z}, \boldsymbol{\theta})$ and the real data, the experiment result is showed in figure 8 and the dash lines in figure 8 represent the mean value of the 100 generators. Interestingly, the result shows that, in each generator, original GAN tends to search a distribution in the distribution range that is closer to the real data distribution, while our algorithm is searching for a distribution that is complementary to the generators' group (i.e., a huge decrease in the mixture condition in figure 7) even if its own performance is poor (i.e., a high distance in the in figure 8).

**CelebA.** We also conduct our experiment on the CelebA dataset Liu et al. (2015). As shown in figure 1, we start with an identical generator and train up to 6 generators using different methods. The measured Wasserstein distance between mixed distribution of Group G and the real-data distribution is showed in figure 1. In this experiment, we use the training method WGAN-GP proposed by Gulrajani et al. (2017). The experiment results indicates that our algorithm outperforms the other two methods after the second generator. It demonstrates the potential of our algorithm applying to real world datasets.

Table 1: Wasserstein distance (WD) between Group G and the CelebA dataset with different numbers of generators (i.e., $|G|$).

| Method | $|G|$=1 | $|G|$=2 | $|G|$=3 | $|G|$=4 | $|G|$=5 | $|G|$=6 |
|---|---|---|---|---|---|---|
| Original GAN | | **588.482** | 337.859 | 307.117 | **189.149** | 146.062 |
| AdaGAN | 789.655 | 607.619 | 569.948 | 322.936 | 206.326 | 248.529 |
| **Incremental GAN** | | 704.004 | **277.993** | **154.544** | 206.693 | **52.9456** |

## 6 FUTURE WORK

We intend to further investigate the problem of how to accelerate the convergence via optimal generator weights, i.e., $\omega_i$. Moreover, by measuring the relativity between the current group G and the dynamically changing target distribution, we intend to apply our algorithm to online learning.

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

## A  MISSING PROOFS

### A.1  PROOF OF LEMMA 2

*Proof of lemma 2.* Without loss of generality, we can assume $\rho = \mathbf{0}$, since otherwise we can add an offset $-\rho$ to the Hilbert space $\mathcal{H}$.

Denote $d_n = n\|T_n\| = \|\sum_{i=1}^{n} p_i^*\|$. Then according to equation $\star$,

$$d_{n+1} = (n+1)\|T_{n+1}\| \leq \inf_{p \in \Pi} \|p + nT_n\| + \epsilon.$$

Let $\hat{p} \in \overline{\Pi}$ be the point that minimizes the distance between $-nT_n$ and its projection $\hat{p}_\perp$ on line $-nT_n$, i.e., $\hat{p} = \arg\min_{p \in \overline{\Pi}} \|p_\perp + nT_n\|$, where $p_\perp = \frac{\langle p, nT_n\rangle_{\mathcal{H}}}{\|nT_n\|^2} \cdot nT_n$.
Then we can further bound $d_{n+1}$ by $\|\hat{p} + nT_n\|$,

$$d_{n+1} - \epsilon \leq \|\hat{p} + nT_n\| = \sqrt{\|\hat{p} - \hat{p}_\perp\|^2 + \|\hat{p}_\perp + nT_n\|^2}.$$

On one hand, since $T_n \in \text{CONV}\,\Pi$, $\hat{p}_\perp$ can be seen as the projection of $\hat{p}$ on $T_n$ as well, hence $\|\hat{p} - \hat{p}_\perp\| \leq \|\hat{p} - T_n\|$. Note that $\Pi$ is bounded, therefore $\|\hat{p} - T_n\|$ is bounded by the diameter of $\Pi$, denoted as $d \geq 0$.

On the other hand, suppose that $p^*$ is the point closest to $\rho = \mathbf{0}$ within $\overline{\text{CONV}\,\Pi}$, i.e., $p^* = \arg\min_{p \in \overline{\text{CONV}\,\Pi}} \|p\|$. Since $\Pi$ is connected, therefore the projection of $\overline{\Pi}$ on $-nT_n$ is the same with the projection of $\overline{\text{CONV}\,\Pi}$. Hence,

$$\|\hat{p}_\perp + nT_n\| \leq \|p_\perp^* + nT_n\| \leq \|p^* + nT_n\| \leq \|p^*\| + d_n.$$

In other words,

$$d_{n+1} - \epsilon \leq \sqrt{d^2 + (\|p^*\| + d_n)^2}.$$

If $\|p^*\| = \inf_{p \in \overline{\text{CONV}\,\Pi}} \|p\| > 0$, then we have $d_n = n\|T_n\| \geq n\|p^*\|$. Hence

$$\begin{aligned}
d_{n+1} &\leq \epsilon + \|p^*\| + d_n + d^2/2(\|p^*\| + d_n) \\
&\leq \epsilon + \|p^*\| + d_n + d^2/2(n+1)\|p^*\| \\
&\leq (\|p^*\| + \epsilon)(n+1) + d^2/2\|p^*\| \cdot \ln(n+1).
\end{aligned}$$

Then for any $\delta > 0$, let $N$ be sufficiently large such that $\frac{\ln N}{N} \leq 2\delta\epsilon\|p^*\|/d^2$, we have

$$\|T_n - \rho\| = d_n/n \leq \inf_{p \in \text{CONV}\,\Pi} \|p - \rho\| + \epsilon + \delta\epsilon.$$

Otherwise, $\|p^*\| = 0$ and $d_{n+1} \leq \epsilon + \sqrt{d^2 + d_n^2}$. Note that the upper bound is increasing in $d_n$, hence $\forall n > 0, d_n \leq d_n^*$ for $d_n^*$ defined as follows:

$$d_0^* = 0, d_{n+1}^* = \epsilon + \sqrt{d^2 + d_n^{*\,2}}.$$

For which, we can easily prove by induction that $d_n^* \leq n\epsilon + \sqrt{n}d$. Therefore $\|T_n\| = d_n \leq \epsilon + d/\sqrt{n}$, which immediately completes the proof. $\square$

### A.2  PROOF OF COROLLARY 1

*Proof of corollary 1.* Without loss of generality, we assume the optimal projection of target distribution is changed from $\rho$ to $\rho'$ after $n_0$ iterations, where $n_0 \in R^+$ is a constant value. Then we can derive $\|T_{n+n_0} - \rho'\| \leq \frac{n \cdot \|T_n - \rho'\|}{n+n_0} + \frac{n_0 \cdot \|T_{n_0} - \rho'\|}{n+n_0}$, where $n \in R^+$ is the training iteration after change.
Then based on lemma 2, we obtain $\lim_{n \to +\infty} \|T_n - \rho'\| \leq \inf_{p \in \Pi} \|p - \rho'\| + \epsilon$. On the other side, for a specific $n_0$, $\|T_{n_0} - \rho'\| \leq \|T_{n_0} - \rho\| + \|\rho - \rho'\|$ is a bounded value if the variation of target distribution is limited.
Finally, we can obtain $\lim_{n \to +\infty} \|T_{n+n_0} - \rho'\| \leq \inf_{p \in \Pi} \|p - \rho'\| + \epsilon$, which concludes the proof. $\square$

## A.3 THE GAP BETWEEN KL DISTANCE AND $\chi^2$ DISTANCE

**Lemma 3.** *At worst case,* $D_{KL}(p||q) \leq D_{\chi^2}(p||q) + 0.42$.

*Proof.* According to f-divergence, we have $D_f(p||q) = \int_x q(x) \cdot f\left(\frac{p(x)}{q(x)}\right) \mathrm{d}x$. For KL-divergence and $\chi^2$-divergence, the corresponding $f(t)$ are $f_{KL}(t) = t\log(t)$ and $f_{\chi^2}(t) = (t-1)^2$ respectively. Import an auxiliary function as:

$$F(t) = f_{\chi^2}(t) - f_{KL}(t)$$

Then based on the monotonicity of F(t), we have $F(t)_{min} \geq -0.42$.

$$
\begin{aligned}
D_{KL}(p||q) &= \int_x p(x) f_{KL}\left(\frac{p(x)}{q(x)}\right) \mathrm{d}x \\
&\leq \int_x p(x) \left(f_{\chi^2}\left(\frac{p(x)}{q(x)}\right) + 0.42\right) \mathrm{d}x \\
&= D_{\chi^2}(p||q) + 0.42
\end{aligned}
$$

which conclude the proof. □

