# OpenReview forum: "Incremental training of multi-generative adversarial networks"
_ICLR.cc/2019/Conference_

### Official Review · AnonReviewer3 · 2018-11-04
**Using multiple generators to properly capture the target (data) distribution**

**Rating:** 6
**Confidence:** 3

**Review:**

In this work, the authors propose to use multiple generators to estimate the target distribution. Especially, it assumes the case that the range of generators is non-convex and the target distribution doesn't fall into it. To solve this issue, the multiple generators are convex combined to do better approximation and an incremental training process is proposed to train multiple generators one by one.

1) Using multiple generators seems reasonable based on the authors assumption (non-convex of the range of generators), but is  this assumption based on having a perfect discriminator? Could you assume a similar case for the discriminator?

2) In figure 3, it is shown each generator tries to improve the estimated target distribution. However, it is not clear what generator generates what samples. It would be better to use different colors for different generators. If I assume that the red samples are from the first generator, why the second image (top right) shows slightly shifted samples compared to the first image (top left). As far as I understand, the first generator is fixed after it is converged.

3) It is shown that the (convex) weights for generators are fixed to 1, is there any reason to fixed it?

4) On page 3, the equation in section 2.1 looks like missing $w_{n+1}$, could you confirm this?

5) is the Original GAN exactly the Ian's original GAN or WGAN?

6) Have you tried this approach using small sized generator (having  small number of parameters)?

---

> ### Author Response · Authors · 2018-11-23
> **Respond to Reviewer3：**
>
> We thank the reviewer for the careful reading and recognizing our contributions
>
> 1. We discuss these problems for multiple times, and we agree that a perfect discriminator can obtain the equality as the lower bound become the virtual ‘distance’. Furthermore, using combination of discriminators can also enlarge the functional space. Another question is how to combine the discriminator results. For discriminators, it needn’t to be a distribution, so we can use affine combination or other combination methods to improve its performance. In a word, it’s a promising work, we would further investigate these theories in the future.
>
> 2. We thank the reviewer for the helpful comments and we redraw the corresponding results in the experiment for clarity. In figure 3, we use different colors as well as corresponding numbers to describe the data points generated by different generators.
>
> 3. Based on the same input distribution and the same architecture for each generator, we assume that the ability of expression for each generator is identical in the target distribution space, so it’s reasonable to set the corresponding weight as the same number. Moreover, in theory, we further prove that when we set the weight to 1, the mixed distribution can converge to the optimal distribution we can get in the convex hull under such specifical settings.
>
> 4. We fix this typo.
>
> 5. The Original GAN means we train each generator using the original target distribution as before, i.e., p_data, and for comparison, the training method we use in the experiment for the three are identical. We rewrite the footnote to make this interpretation more clearly.
>
> 6. We tried this approach under a small sized condition. Heuristically, a small sized generator means a weak fitting capacity, and in our experiments, due to the continuous latent variable, a simple architecture may generate more samples fall into the clearance between different modes, thus obtain a worse performance.

---

### Official Review · AnonReviewer1 · 2018-11-08
**A sequential training of GANs and some theory associated with it**

**Rating:** 6
**Confidence:** 4

**Review:**

This paper proposes a method for ensembling GAN’s for capturing diversity in the target space. This done by a convex combination of GAN’s that are sequentially trained by trying to approximate the real distribution by fixing the previous generators. The paper theoretically shows that this approach converges to the optimal theoretical distribution.

Comments

1)  What will be the performance of Original GAN and Incremental GAN by finding optimal weighting ($w_i$) parameters for each of them?
2)  Can you increase the number of parameters of the generator by no of generators used in the incremental GAN’s and compare the performance?
3) The abstract first line you have written ‘possibly distribution’ instead of ‘probability distribution’
4) Table 1 ‘Incremental GAN’ doesn’t show consistently improved performance in comparison ‘Original GAN’. Can you train a few more generators and verify it?

---

> ### Author Response · Authors · 2018-11-23
> **Respond to Reviewer1**
>
> We thank the reviewer for the careful reading and insightful comments.
>
> 1. It’s a promising future work to find the optimal weight (i.e., w_i) for each generator, and this approach can accelerate the convergence. E.g., one can imagine a special case that the range set is a minor arc lies in a plane and the target points lies on the chord corresponding to the arc (just like the sketch map below).
>
> ___A
> |
> |
> | C  . p_data
> |
> |___B
>
> The optimal combination is easy to get: w_A * A+w_B * B. But using a greedy algorithm, we may first find the solution that is nearest to p_data, i.e., C. After that, the target distribution for next generator is shifting to 2*p_data – C, a point lies on the reverse extension line of line p_data-C. After several iterations, we can find the point A and B as the solution for the generator (At this time, A and B is the nearest point to the target point n*p_data-(n-1)*p_pre.). In this extreme case, we need to repeat infinite times to offset the horizontal shifting, while if we can optimize the weights, the optimal combination can be obtained in the finite iterations.
>
> 2. Increase the number of parameters may enlarge the distribution range of each generator, and thus we could get a better solution to approximate real data distribution through a single generator. But these increments may change the distribution range for each generator, however, in each iteration, we are indeed in search of the solution in the different distribution sets, so in theory, we can’t guarantee the combination of the generators is better than before.
>
> 3. We fix this typo.
>
> 4. We further train up to 10 generators to approximate the real data distribution, and we didn’t project the data to the interval (-1, 1), so the difference is more significant, the results show below:
> ---------------------------------------------------------------------------------------------------------------------------------------
>    Method         |G|=1      |G|=2     |G|=3      |G|=4      |G|=5      |G|=6     |G|=7
> ----------------------------------------------------------------------------------------------------------------------------------------
> Incremental GAN  52003.589  7655.897  7494.769   14156.5    4016.2288  2600.899  23281.205
> Original GAN   52003.589   57966.67  27102.465  32559.604  31377.982  31643.3   30829.93
> ----------------------------------------------------------------------------------------------------------------------------------------
>
> ----------------------------------------------------------------------
>    Method         |G|=8      |G|=9     |G|=10
> ----------------------------------------------------------------------
> Incremental GAN  17003.852   9159.07    9118.682
> Original GAN    35698.953   32310.938  44249.152
> ----------------------------------------------------------------------
> The oscillation is because the generator converges to some local optimal, thus influence the whole performance. Furthermore, our algorithm can fast recover from the deviation, and get closer to it again in the successive iterations.

---

### Official Review · AnonReviewer4 · 2018-11-13
**Incremental training of GANs**

**Rating:** 5
**Confidence:** 3

**Review:**

The paper introduces an incremental training method for GAN's for capturing the diversity of the input space. The paper demonstrate that the proposed method allows smaller distances between the true and generated distribution. I find the idea interesting, but fear that the 60-100 small ensemble models could be replaced by a larger model.

I am curious about why we need incremental training when it seems like we could directly train all the networks jointly. The corresponding generative model is simply stronger so all the convergence arguments would still hold. Is the statistical distance a reasonable estimate for you to determine whether you need an additional generator for incremental training?

Also what are the generator architectures for the experiments? How can you put 60-100 generators within the GPU memory? The latent variable dimension seem to be only 1 for each of your generator? That seems to be seriously handicapping the capacity of each individual generator (to just some data points), so the ensemble distribution might be obtained simply by using a larger dimension z?

There are also other measurements that are used by the GAN community, such as inception score, FID score and samples. It seems also reasonable to verify the effectiveness of this method on CIFAR or LSUN datasets, where the method would have a greater improvement because the data distributions are more complex.

Minor points:
- How do you measure the "Wasserstein distance" for high-dimensional distributions?
- What not set $\omega_i$ to be always 1? The subsampling process introduced in Algorithm 2 seem to enforce this, and you do this for all the experiments.
- Fix citation typos.
- Fix \mathbf for vector quantities, such as x and z.
- Since the generative models have the same architecture, does the non-convex argument becomes moot when you have a mixture of 2 generators?

---

> ### Author Response · Authors · 2018-11-23
> **Respond to Reviewer4**
>
> We thank the reviewer for careful reading and helpful comments.
>
> 1. We propose incremental training because of its flexible. A jointly training algorithm is limited by its number of components. As [1] shows, due to the limit of GPU memory, they can only get mix+GAN with components T<=5. As in our algorithm, we don’t need to store all the parameters in the memory. In our new loss function, we could store a sets consisted of samples generated from each generator, and then get batches from these sample sets to approximate the expectation of each generator, i.e., E_{x~g_i}[D(x)]. We argue that we could also add the generators group by group using the jointly training algorithm to search for a complementary mixture distribution instead of one by one, and these may not be limited by the GPU memory.
>
> 2. We use the architectures resemble to DC-GAN in all our experiments except the Gaussian mixture sets experiments. In Gaussian mixture experiment, we use generators contain 2 hidden layers of size 128 with a relu activation fully connected network, followed by a linear projection to 2 dimensions.
> For the second question, we don’t need to put all generators within the GPU memory, after i-th iteration, we store a new folder consisted of samples generated by the new generator, and make a pipeline to read the batches to approximate the expectations of E_{x~g_i}[D(x)], so we could add multiple generators into the generator group.
> We reduce the latent variable to dimensional 1 so that the condition of figure 1(c) is likely met, and we also test our algorithm by using a lager dimension z, the only difference between different dimensions is that a lager dimension z can get a smaller distance of start point, and the convergence distance is also smaller than a small dimension z, so we could also obtain a better solution through a combination of generator using the larger dimension z.
>
> 3. We are trying to implement these experiments and we will measure our method on CIFAR and LSUN to test the performance of our algorithm. We will give the results soon.
>
> Minor points:
> 1. Through the WGAN, the loss function W(P_r, P_g)=sup_{||f||_L<=1}{E_{x~P_r}[f(x)]-E{x~P_g}[f(x)]} is a elegant lower bound to "Wasserstein distance", when we fix the input data distribution for both the real data and the generated data, then train an auxiliary discriminator to maximize the loss function, we can get an approximate "Wasserstein distance".
>
> 2. Based on the same input distribution and the same architecture for each generator, we assume that the ability of expression for each generator is identical in the target distribution space, so it’s reasonable to set the corresponding weight as the same number. Moreover, in theory, we further prove that when we set the weight to 1, the mixed distribution can converge to the optimal distribution we can get in the convex hull under such specifical settings.
>
> 3. We fix these typos.
>
> 4. We fix these expressions.
>
> 5. Thank you for your thorough thinking, we think this argument is meaningful. First, consider a set consisted of the endpoints of a simplex, under this condition, a mixture of 2 endpoints can only get the point on the boundary, and we should make combination of all endpoints to get each point within the simplex. Since the distribution range is complex, a mixture of 2 generators may not enough to get the optimal one in the convex hull.
>
> ____________________________________________________________________________________________
>
> [1]. Sanjeev Arora, Rong Ge, Yingyu Liang, Tengyu Ma, and Yi Zhang. Generalization and equilibrium in generative adversarial nets (gans). In Doina Precup and Yee Whye Teh (eds.), Proceedings of the 34th International Conference on Machine Learning, ICML 2017, Sydney, NSW, Australia, 6-11 August 2017, volume 70 of Proceedings of Machine Learning Research, pp. 224–232. PMLR, 2017.

---

### Meta-Review · Area_Chair1 · 2018-12-18

**Confidence:** 4
**Recommendation:** Reject

**Metareview:**

The reviewers and the AC acknowledge the paper contains interesting ideas on using an incremental sequence of multiple generators to capture the diversity of the examples. However, the reviewers and the AC also note that the potential drawback of the paper is the lack of evaluation with other metrics such as inception score, FID score, etc. Therefore the paper is not quite ready for acceptance right now, but the AC encourages the authors to submit to other top venues with more thorough experiments.